Post-transcriptional regulation of several biological processes involved in latex production in Hevea brasiliensis

Leclercq Julie 1 2
Wu Shuangyang 3
Farinas Benoît 1 2
Pointet Stéphanie 1 2
Favreau Bénédicte 1 2
Vignes Hélène 1 2
Kuswanhadi Kuswanhadi 4
Ortega-Abboud Enrique 1 2
Dufayard Jean-François 1 2
Gao Shenghan 3
Droc Gaëtan 1 2
Hu Songnian 3
Tang Chaorong 5
Montoro Pascal pascal.montoro@cirad.fr 1 2
1 CIRAD, UMR AGAP , Montpellier , France
2 AGAP, University of Montpellier, CIRAD, INRAE, Institut Agro , Montpellier , France
3 University of Chinese Academy of Sciences, CAS Key Laboratory of Genome Sciences and Information, Beijing Institute of Genomics, Chinese Academy of Sciences , Beijing , China
4 Sembawa Research Centre, Indonesian Rubber Research Institute , Palembang , Indonesia
5 Hainan University, College of Tropical Crops , Haikou , China
Lazo Gerard
Electronic publication date: 2020 Apr 29
Publication date: 2020
Volume: 8
Electronic Location ID: e8932
Received 2019 Nov 7; Accepted 2020 Mar 17
Copyright: ©2020 Leclercq et al.
Copyright year: 2020
Copyright holder: Leclercq et al.
License: This is an open access article distributed under the terms of the Creative Commons Attribution License, which permits unrestricted use, distribution, reproduction and adaptation in any medium and for any purpose provided that it is properly attributed. For attribution, the original author(s), title, publication source (PeerJ) and either DOI or URL of the article must be cited.
License URL: https://creativecommons.org/licenses/by/4.0/

Keywords: Small RNA, miRNA, Degradome, Rubber tree, Crop epigenomics, Abiotic stress, Latex

Funding: UCAS Joint PhD Training Program Campus France 41241QJ Shuangyang Wu benefited a scholarship from the UCAS Joint PhD Training Program. Julie Leclercq and Shuangyang Wu benefited a mobility program in 2018 from Campus France (Xu Guangqi Program no 41241QJ). The funders had no role in study design, data collection and analysis, decision to publish, or preparation of the manuscript.

==============================
Background

Small RNAs modulate plant gene expression at both the transcriptional and post-transcriptional level, mostly through the induction of either targeted DNA methylation or transcript cleavage, respectively. Small RNA networks are involved in specific plant developmental processes, in signaling pathways triggered by various abiotic stresses and in interactions between the plant and viral and non-viral pathogens. They are also involved in silencing maintenance of transposable elements and endogenous viral elements. Alteration in small RNA production in response to various environmental stresses can affect all the above-mentioned processes. In rubber trees, changes observed in small RNA populations in response to trees affected by tapping panel dryness, in comparison to healthy ones, suggest a shift from a transcriptional to a post-transcriptional regulatory pathway. This is the first attempt to characterise small RNAs involved in post-transcriptional silencing and their target transcripts in Hevea.

Methods

Genes producing microRNAs (MIR genes) and loci producing trans-activated small interfering RNA (ta-siRNA) were identified in the clone PB 260 re-sequenced genome. Degradome libraries were constructed with a pool of total RNA from six different Hevea tissues in stressed and non-stressed plants. The analysis of cleaved RNA data, associated with genomics and transcriptomics data, led to the identification of transcripts that are affected by 20–22 nt small RNA-mediated post-transcriptional regulation. A detailed analysis was carried out on gene families related to latex production and in response to growth regulators.

Results

Compared to other tissues, latex cells had a higher proportion of transcript cleavage activity mediated by miRNAs and ta-siRNAs. Post-transcriptional regulation was also observed at each step of the natural rubber biosynthesis pathway. Among the genes involved in the miRNA biogenesis pathway, our analyses showed that all of them are expressed in latex. Using phylogenetic analyses, we show that both the Argonaute and Dicer-like gene families recently underwent expansion. Overall, our study underlines the fact that important biological pathways, including hormonal signalling and rubber biosynthesis, are subject to post-transcriptional silencing in laticifers.

Introduction

Small RNAs (sRNAs) modulate plant gene expression at both transcriptional and post-transcriptional levels (Bazin & Bailey-Serres, 2015; Khraiwesh, Zhu & Zhu, 2012; Wang et al., 2016). They include micro-RNAs (miRNAs) and small interfering RNAs (siRNAs), originating from single-stranded hairpin or double-stranded RNA precursors, respectively (Bologna & Voinnet, 2014). MiRNAs are preferentially expressed from MIR genes, and consist of 20–22 nucleotides (nt) sRNAs. They are known to regulate gene expression through either cleavage and degradation or translational repression of target mRNAs. Small RNAs of 23–25 nt may either target their parent transcripts, or trigger RNA-directed DNA methylation (RdDM) of the corresponding genes (Bologna & Voinnet, 2014).

The proteins involved in biogenesis are highly diversified among plant species, which may explain the emergence of new functions and regulatory pathways (D’Ario, Griffiths-Jones & Kim, 2017). The impact of sRNA-mediated regulations on plant phenotypes has been illustrated through several examples, including association between increased apple fruit size and MIR172 gene deregulation (Yao et al., 2016), and alleviation of siRNA-dependent repression of FAD2 expression resulting in increased oil content in olive fruits (Unver et al., 2017). In annual plant species, size classes and respective abundances of sRNA transcriptomes may be affected by a variety of environmental stresses (Ferdous, Hussain & Shi, 2015; Khraiwesh, Zhu & Zhu, 2012; Liu et al., 2017; Yang et al., 2017; Zhao et al., 2016). Similar observations have been made in cultivated perennial species (Beule et al., 2015; Li et al., 2017).

Previous work by our team on Hevea brasiliensis revealed alterations in the sRNA transcriptome of latex cells in response to environmental cues (Gébelin et al., 2013b). This tropical perennial crop is used for the production of latex containing natural rubber (cis-1,4 polyisoprene) (Compagnon et al., 1986). Severe environmental constraints and stresses associated with latex harvesting practices, such as wounding through tapping or ethephon stimulation, cause physiological disorders leading to tapping panel dryness (TPD). TPD is characterized by a halt in latex flow and in situ latex coagulation (for a review Zhang, Leclercq & Montoro, 2017). In a previous study comparing healthy and TPD-affected trees, we observed a shift in the respective proportions of different size classes of sRNA in latex (Gébelin et al., 2012; Pramoolkit et al., 2014), the 24 nt size class predominating in healthy trees, and the 21 nt size class in TPD-affected ones. This observation led to the need to annotate small RNAs to decipher their biological activity, in particular that of 20–22 nt (miRNA and ta-siRNA). Although not complete, previous studies characterised endogenous miRNAs in Hevea, thereby enabling the identification of 68 and 16 species-specific families, thanks to the availability of a reference transcriptome from clone PB 260 (Gébelin et al., 2012; Gébelin et al., 2013b; Kanjanawattanawong et al., 2014; Lertpanyasampatha et al., 2012). The aim of the present study was to complete the annotation of Hevea microRNAs and ta-siRNAs thanks to the availability of genomic sequences from clone PB 260 (Zhang et al., 2019). Previous computational prediction of miRNA targets led to the identification of 1,788 sequences (Gébelin et al., 2012; Pramoolkit et al., 2014). However, among these target miRNAs, few have been experimentally validated so far (Gébelin et al., 2012; Pramoolkit et al., 2014). Therefore, the real impact of rubber tree miRNAs on transcript regulation remains to be fully assessed.

Unlike animal miRNAs, the main mode of action of plant miRNAs is through transcript cleavage by endonucleolytic slicing (Arribas-Hernandez, Kielpinski & Brodersen, 2016), thereby generating degradation products that are collectively known as “degradome”. To improve our knowledge of post-transcriptional regulatory mechanisms in the clone PB 260 rubber tree, we generated and analysed six degradome libraries originating from six different tissues to detect miRNA- and ta-siRNA-mediated cleaved targets. Their 3′ ends were then sequenced so that all sRNA-directed cleavage sites could be identified simultaneously (German et al., 2009). The targets were then functionally annotated.

Here we discuss our results with a focus on the laticiferous tissue and the natural rubber biosynthetic pathway. Our results suggest that laticiferous tissues have the highest level of post-transcriptional regulation by mRNA cleavage, and that the natural rubber (NR) biosynthetic pathway is under strong post-transcriptional silencing. These results could be used to achieve functional validation in rubber trees, under the physiological stresses associated with latex harvesting.

Materials & Methods

Plant material and RNA samples

The plant material used for this study comes from multiple origins but all from Hevea clone PB 260. Samples were collected during in-vitro culture, from in-vitro plants, and from juvenile budded plants grown in the greenhouse at CIRAD (Montpellier). Other samples were collected from mature trees grown in the field at the Sembawa Rubber Research Center in Indonesia and the Thailand Chachoengsao Rubber Research Centre.

Total leaf, bark and root RNAs were generated from juvenile plants subject to cold, high light, drought, flooding, salinity and wounding stress by Gébelin et al. (2012). Total leaf and bark RNAs from juvenile plant treated with hormones (ethylene, methyl jasmonate) were generated by Duan et al. (2010) and Kuswanhadi et al. (2010). Total RNAs from latex and bark of healthy and TPD affected mature trees were prepared by Gébelin et al. (2013b), and total RNAs from male and female flowers by Piyatrakul et al. (2012). Total RNAs from somatic embryos were generated by Piyatrakul et al. (2012). All these total RNAs from in-vitro culture, juvenile and mature plants were used to build specific leaf, bark, root, embryo, flower and latex degradome libraries.

Available data sets from five small-RNA-seq data were generated by (Gébelin et al., 2012; Gébelin et al., 2013b; Kanjanawattanawong et al., 2014; Lertpanyasampatha et al., 2012) comprising juvenile plants, latex of healthy and TPD-affected trees, young and fully developed leaves from mature trees. Complete tissue-specific (embryo, flower, latex, leaf, bark, root) transcriptomes were generated by Duan et al. (2013) and Piyatrakul et al. (2012).

Differential expression analysis of RNAseq data in latex from mature trees were performed by Montoro et al. (2018). Pairwise comparisons were performed between untreated and ethephon-treated healthy trees (WH vs EH) and between healthy and TPD ethephon-treated trees (EH vs ET).

Nuclear genome resequencing of clone PB 260

Leaf nuclear DNAs from a budded clone PB 260 were sequenced by GATC (https://www.eurofinsgenomics.eu/) using Illumina pair-end sequencing (2 × 150 bp). About 280 million reads were generated, which correspond to a total of 84 Gb among which 62 Gb were assembled through alignment against the clone Reyan7-33-97 reference genome (Tang et al., 2016). The Bowtie tool (Langmead et al., 2009) was used to build a Burrows-Wheeler transform index to map small RNA libraries with specific parameters [bowti e − f − v 1 -p 10 − a − m 50 –best –strata (Langmead et al., 2009)]. The statistics used to resequence the genome of clone PB 260 are detailed in (Zhang et al., 2019). The resequencing data are available under project number PRJCA001333 in the GSA (Wang et al., 2017a) and BIG Data Center (BIG Data Center Members, 2018).

MIR gene identification

MIR genes were identified on the PB260 resequenced genome. MITP (https://sourceforge.net/projects/mitp/) was chosen with default parameters to identify hairpin structures and miRNAs in available small RNA datasets (Gébelin et al., 2012; Gébelin et al., 2013b; Kanjanawattanawong et al., 2014; Lertpanyasampatha et al., 2012). The small RNA–seq data described in Gébelin et al. are available under project number PRJCA001333 in the GSA (Wang et al., 2017a) and BIG Data Center (BIG Data Center Members, 2018). MITP uses RNAfold in the Vienna RNA Package (Piyatrakul et al., 2014) to detect hairpin structures. The longest hairpin sequences were further annotated using BLAST (Altschul et al., 1990) against MirBase (Kozomara & Griffiths-Jones, 2011) and plant non-coding RNA database (PNRD) (Yi et al., 2015) as references (Fig. 1).

Figure 1 Scheme representing the analysis process fordeciphering post-transcriptional regulations in Hevea by miRNAs.

Each step is commented on the ‘Results’ section. The pipelines or tools used for the detection, classification and functional identification of miRNAs and their targets are shown in italic styling.

Degradome libraries

Degradome libraries were constructed from a pool of total RNA from stressed and non-stressed plants, representing six different Hevea tissues (leaf, root, latex, bark, embryos and reproductive tissues) (see plant material), according to the protocol described in (German et al., 2009), with major modifications to account for the different sequencing techniques used (Miseq versus GAII in the 2009 publication). Briefly, libraries were produced from 150 µg of total RNA. The 5′ RNA-adapter sequence is RA5: 5′-GUUCAGAGUUCUACAGUCCGACGAUC- 3′. The 3′-adapter sequence is 5′-GTGACTGGAGTTCCTTGGCACCCGAGAATTCCATTTTTTTTTTTTTTTTTTV- 3′. The primer sequences for library amplification are 5′ Adapter Primer (RP1: 5′ AATGATACGGCGACCACCGAGATCTACACGTTCAGAGTTCTACAGTCCGA- 3′) with the 3′ Primer Adapter indexed (RPI1: 5′-CAAGCAGAAGACGGCATACGAGAT CGTGATGTGACTGGAGTTCCTTGGCACCCGAGAATTCCA- 3′, RPI2: 5′-CAAGCAGAA GACGGCATACGAGATACATCG GTGACTGGAGTTCCTTGGCACCCGAGAATTCC A-3′, RPI3: 5′-CAAGCAGAAGACGGCATACGAGATGCCTAA GTGACTGGAGTTCCT TGGCACCCGAGAATTCCA-3′, RPI4: 5′-CAAGCAGAAGACGGCATACGAGATTGGT CA GTGACTGGAGTTCCTTGGCACCCGAGAATTCCA-3′, RPI5: 5′-CAAGCAGAAGA CGGCATACGAGATCACTGT GTGACTGGAGTTCCTTGGCACCCGAGAATTCCA-3′, and RPI6: 5′-CAAGCAGAAGACGGCATACGAGATATTGGC GTGACTGGAGTTCCTT GGCACCCGAGAATTCCA-3′). The degradome data are available under project number PRJCA001333 in the GSA (Wang et al., 2017a) and BIG Data Center (BIG Data Center Members, 2018). The number of sequences produced and statistics are listed in Table S1.

Detection and GO classification of cleaved target transcripts

Raw sequencing reads originating from degradome libraries were trimmed with Cutadapt (Martin, 2011). The CleaveLand pipeline (Addo-Quaye, Miller & Axtell, 2009) was used in Mode 1 (the mode that allows alignment of degradome data, creation of degradome density file, and new small RNA query/transcriptome alignment with GSTAr) in order to identify sRNA-generated transcript cleavage products (Fig. 2).

Figure 2 Scheme representing the exploration of degradome libraries.

CleaveLand pipeline is used in order to detect miRNA-mediated cleaved targets, with transcriptome and small RNA dataset as input. The abundance of reds starting at the cleavage site is define by the degradome category (see ‘Materials & Methods’).

The abundance of reads starting at the cleavage site is defined by the degradome category. Category 4 has just one read at this position. Other categories have more than one read. In category 3, the number of reads is below or equal to the average depth of coverage (all positions that have at least one read) on the transcript. In category 2, the number of reads is equal to the average depth of coverage on the transcript. In category 1, the number of reads is equal to the maximum of the average depth of coverage on the transcript when there is more than one position at the maximum value. In category 0, the number of reads is equal to the maximum of the average depth of coverage on the transcript when there is just one position at the maximum value. The outputs are presented in Tables S6, S9–S13 for miRNAs and S15, S17-21 for ta-siRNA).

Functional annotation of the identified targets was performed using InterProScan (v5.18–57.0) in STANDALONE mode. GO classification was then extracted from the resulting tabular output (Ashburner et al., 2000).

Prediction of trans-acting siRNA production sites

Small RNAs were aligned to the PB 260 genome assembly, and small RNAs that did not match were discarded. As input sRNA files, we used trimmed reads from latex cells from healthy and TPD-affected trees and juvenile plants, with a minimum sRNA abundance of two, a p-value cut-off of 0.0001 and a length of 21 nt. Using the small RNA workbench from the University of East Anglia (http://srna-workbench.cmp.uea.ac.uk/; version 4.3.1 Alpha) (Stocks et al., 2012), the 21-mers ta-siRNA loci were identified using the TA-SI Prediction tool (p-value cut-off <0.0001). The algorithm for ta-siRNA enabled us to calculate phasing probability based on hypergeometric distribution (Chen, Li & Wu, 2007). Coding potential was estimated at each of the identified sites using the Coding Potential Calculator (CPC) (http://cpc.gao-lab.org/) (Fig. 3).

Figure 3 Scheme representing the analyses process for deciphering post-transcriptional regulations in Hevea by ta-siRNAs.

Each step is commented on the ‘Results’ section. The pipelines or tools used for the detection, classification and functional identification of ta-siRNAs and their targets are shown in italic styling.

Annotation of the miRNA biogenesis and regulatory pathway and phylogenetic analysis

A list of 95 Arabidopsis proteins involved in the RNA-mediated silencing machinery was extracted from the TAIR database (https://www.arabidopsis.org/) (Matzke & Mosher, 2014). The amino acid sequences were used to retrieve putative orthologous sequences using the SPALN pipeline with the parameters -O0 -Q4 -M5 -H180 (Iwata & Gotoh, 2012), from the PB 260 genomic sequences and the reference Hevea brasiliensis transcriptome (Piyatrakul et al., 2012). A set of 206 genes encoding putative AGO proteins from 23 plant species, and 75 genes and putative DCL proteins from 17 plant species, was used to locate candidate orthologous sequences in the rubber tree genome using the SPALN pipeline. The presence of signatures corresponding to conserved domains for either of these protein families, such as Piwi (PF02171), PAZ (PF02170) and Dicer dimerization (PF03368), was ascertained in the filtered genes using Interproscan. Putative H. brasiliensis AGO and DCL peptide sequences were classified through alignment against the corresponding full-length amino acid sequences from Populus trichocarpa, Ricinus communis and Arabidopsis thaliana (Daniel Rodríguez-Leal et al., 2016; Mukherjee, Campos & Kolaczkowski, 2013) using MUSCLE (version 3.8.31) (Edgar, 2004a; Edgar, 2004b). Cleaning was performed to remove gaps using trimAl (version 1.4) (Capella-Gutierrez, Silla-Martinez & Gabaldon, 2009), after which a phylogenetic tree was built using MrBayes (v3.2.6 x64).

Results

The analyses of miRNAs and of degradomes from data processing to functional annotation by gene ontology are depicted in Fig. 1. The first step was to annotate the stem-loop structures and the resulting miRNAs. The second step was to analyse the distribution of predicted miRNAs according to size by comparing three conditions in juvenile plants, and latex from healthy and TPD-affected trees. The third step was to analyse their cleavage activities either using the tissue-specific transcriptomes or a list of well-characterised rubber tree genes. Figure 3 shows the process of annotation of ta-siRNAs and their cleavage activities.

Annotation of MIR genes forming RNA hairpin structures and producing small RNAs

Hevea miRNAs and MIR genes (hairpin-forming sequences) were annotated based on high-throughput sequencing data obtained from the five sRNA libraries and from the resequenced genome of clone PB 260. The genome was mapped on the reference genome of clone Reyan7-33-97 (Table S1). An atlas of 1,042 MIR genes, associated with miRNA production, was generated (Fig. 1, Tables S2 and S3), and annotated.

Using both the MirBase and PNRD databases, only, respectively, 29 and 36 MIR sequences matched known MIR sequences from different plant species (Table S4). This study complements our previous analyses (Gébelin et al., 2012; Gébelin et al., 2013b), which showed only eight conserved MIR genes annotated from juvenile plants, and four conserved and 11 new MIR genes from latex of healthy and TPD-affected trees.

To perform a degradome analysis, Ahmed and co-workers (Ahmed et al., 2014) recommend taking miRNA and isomiRNA (miRNA variants) pools into account. A Hevea sRNAome containing 1,839 redundant sequences was constructed with 20–22 nt sized sRNA sequences originating from MIR genes, with or without similarity to known miRNAs (Fig. 1, Table S5). This file was used as input to compare miRNA size class distribution across sRNA libraries, and to explore the degradome libraries to detect cleaved target transcripts/miRNA pairs.

Determination of miRNA size class distribution and base preference

The distribution of unique miRNA accessions identified in small RNA libraries from juvenile stressed plants, latex from healthy and TPD-affected tree tissues is shown in Fig. 4. The total lengths ranged from 17 to 35 nucleotides. The distribution of the size classes showed a major peak at 21 nt, which accounts for 24.4 to 25.2% of the total number of predicted miRNAs (Fig. 4). Interestingly, this proportion was seen to be globally conserved in the different libraries tested (Fig. 1, Tables S2 and S3). The 5′ nucleotide identity of a miRNA guides the loading into AGO proteins (Rogers & Chen, 2012) (Frank et al., 2012; Zha, Xia & Yuan, 2012; Yu, Jia & Chen, 2017). AGO1 is known to sort miRNA with a 5′ uracil, and AGO2, 4, 7 and 10 also showed slicing activity with no sorting priorities (Rogers & Chen, 2012). Hevea predicted miRNAs 21-nt in size were further analysed for their 5′ base preference (Fig. 5). In the latex from healthy and TPD-affected trees, the 5′ nucleotide of predicted miRNA was mainly uracil, in contrast to the guanidine found in the latex from juvenile plants.

Figure 4 Length distribution of unique miRNA accessions.

MiRNAs were annotated by the MITP pipeline for Hevea brasiliensis plant tissues leaf, bark, root (black, Gébelin et al., 2012) and latex from healthy (dark grey) and TPD-affected (light grey) trees (Gébelin et al., 2013b).

Figure 5 Five prime base preference for small RNA of 20–22 nt in each small RNA data set.

Each base preference is indicated by a specific color.

Detection of 136 cleaved target transcript/miRNAs pairs in the degradome at the transcriptome level

The sequencing of the six degradome libraries, from six distinct tissues (leaf, bark, root, flowers, latex and embryos), generated a total of 16 million single reads. The highest and lowest numbers of cleaned reads were found in latex (134,611 reads) and embryo (9,283 reads) libraries, respectively. The distribution of the number of reads per library after the cleaning process is shown in Table S1.

To validate the quality of the degradome libraries, we looked for the position of miRNA-directed cleavage sites for three known target transcripts that had previously been identified experimentally (Gébelin et al., 2012). We successfully detected SBP/CL2120Contig2 targeted by miR156/acc_480780, ARF/CL6582Contig1 by miR160/acc_370 and chloroplastic CuZnSOD/CL4308Contig 2 by miR398/acc_420 (Fig. 6). Degradome products corresponding to these known targets were detected in degradome libraries only in certain tissues, suggesting post-transcriptional regulation could be tissue specific (Table S6). For instance, cleavage products of the chloroplastic CuZnSOD were found in latex and root libraries. Those of ARF were identified in libraries from bark, and those of SBP in leaf and bark. Moreover, we checked for the statistical classification of the category of degradome, which is based on coverage at the cleavage site compared to the total length of the transcript. The three known miRNA-mediated cleaved targets belonged to the worst degradome, i.e., category 4, consisting of only 1 read at the cleavage site (Fig. 6). This result reveals that even miRNA-mediated cleaved targets predicted in weak degradome category can be biologically active.

Figure 6 T-plots generated by CleaveLand.

Alignments show the cleavage site experimentally validated (Gébelin et al., 2012), corresponding exactly to the cleavage site predicted by the degradome analysis showed by a red point on the T-plot. (A) SBP (CL2120Contig2)/miR156 (acc_480780). (B) ARF (CL6582Contig1)/miR160 (acc_370). (C) Chloroplastic CuZnSOD (CL4308Contig 2)/miR398 (acc_420).

At the transcriptome level, our analyses of the degradome products highlighted the prevalence of post-transcriptional regulatory events in latex cells and reproductive tissues in Hevea brasiliensis. Among the 136 transcript targets detected among all the tissues, 39 (28.7%) and 38 (27.9%) were detected in latex and reproductive tissues, respectively (Fig. 1, Table 1, Table S7). By contrast, the embryo degradome yielded the lowest number of targets with only 4 (2.9%) detected. Only 26 target transcripts (19.1%) were cleaved by known miRNAs (16 different accessions). The miRNAs involved in mRNA slicing exhibited a tissue-specific activity pattern with very little overlap between tissues, as shown in the Venn diagram (Fig. 7).

Table 1 Number of identified cleaved targets by miRNAs and functional annotation by GO in the tissue-specific transcriptomes.

Tissue-specific transcriptome/ Degradome libraries	Number of targets	Biological process	
Latex	39	Proteolysis, nitrogen compound metabolic process, response to stress, metabolic process, ATP synthesis coupled proton transport, regulation of RNA metabolic process	
Root	11	Protein phosphorylation, cation transport, cell cycle, signal transduction	
Leaf	31	Carbohydrate metabolic process, glycolytic process, DNA methylation, transcription, DNA-templated, regulation of transcription, DNA-templated, translation, protein phosphorylation, proteolysis, photosynthesis	
Bark	13	Carbohydrate metabolic process,(1->3)-beta-D-glucan biosynthetic process, glycolytic process, glutamine biosynthetic process, nitrogen compound metabolic process, signal transduction, photosystem II assembly, malate transport, photosynthesis, protein import	
Embryos	4	–	
Reproductive tissues	38	Regulation of transcription, DNA-templated, translation, photosynthesis, oxidation–reduction process	

Figure 7 Venn diagram built with sRNA accessions withslicing activity.

Six tissue-derived degradome libraries were tested (leaf, bark, root, reproductive tissues, latex, embryo).

All the cleaved targets identified belonged to degradome categories 3 and 4 (meaning only one or very few reads starting at the cleavage site), except for six mRNAs which were in degradome category 0 (meaning the majority of reads started at the cleavage site). Among them, functional domains were predicted in the putative peptides of only two of the targets found in latex, CL5497Contig1, which included a leucine-rich repeat domain (cysteine-containing subtype), and CL8280Contig1, which included an RNA-binding/CRM domain. Gene ontology annotation strongly suggests that a wide range of biological processes are under post-transcriptional regulation in different rubber tree tissues, including sugar biosynthesis in leaves, ion transport in roots, protein imports and sugar metabolism in the bark, and RNA metabolic processes in latex (Fig. 1, Table 1, Table S8).

Detection of 173 additional miRNA-mediated cleaved transcripts in curated genes involved in laticifer differentiation and natural rubber production

The occurrence of post-transcriptional regulation of genes involved in important biological pathways of Hevea was then investigated in more detail. The gene families have already been described by Duan et al. (2013), Kuswanhadi et al. (2010), Pirrello et al. (2014), Piyatrakul et al. (2012), Putranto et al. (2015a), Putranto et al. (2015b), Tang et al. (2016), Venkatachalam, Thulaseedharan & Raghothama (2007) and Venkatachalam, Thulaseedharan & Raghothama (2009) (Fig. 1, Table 2, Tables S9 to S13). In the present study, a high proportion of transcripts were found to be cleaved by miRNA, including genes with functions related to ethylene biosynthesis (23 genes out of 34 tested including members of the ACO, ACS, ETR, CTR and EIN3 gene families), jasmonate biosynthesis (30 out of 50 genes tested, including a MYC transcription factor), and hormone signalling pathways (54 out of 149 including HbERF-IXb02; HbERF-VIIa07; HbERF-VIIa2; HbERF-VI05 HbERF-VIIa8, HbRAV-03; HbERF-VIIa5). In addition, among the 596 genes selected for their likely involvement in rubber biosynthesis pathway (Tang et al., 2016), 56 had degradation products compatible with post-transcriptional regulation (HbMVK1,2,3; HbMVD1,2; HbREF3,7,8; HbSRPP2; HbACAT3; HbHMGR1; HbHMGS1,2; HbPMD2 and HbGSP3 genes).

Table 2 Identified targets cleaved by miRNA (with redundancy) from gene sequence lists in the degradome libraries.

Gene list/Degradome libraries	Latex	Root	Leaf	Bark	Embryos	Reproductive tissues	
NR biosynthesis	18	10	13	29	3	6	
AP2/ERF super family	19	6	10	8	0	11	
JA biosynthesis and signalling	5	3	10	5	3	6	
TPD-related genes	1	2	2	3	0	1	
ET biosynthesis and signalling	8	7	15	20	1	6	

With respect to TPD-associated genes (Venkatachalam, Thulaseedharan & Raghothama, 2007; Venkatachalam, Thulaseedharan & Raghothama, 2009), nine out of 12 genes showed evidence of post-transcriptional regulation in all tissues tested, with the exception of embryo. Among them, transcripts produced by HbTOM20, which encoded a translocase of the outer mitochondrial membrane, were cleaved by both miR396 and miR395. Moreover, a gene similar to HbTCTP, which has multiple functions and is associated with stress/hormone responses and TPD, may be cleaved by two newly identified miRNAs (Pyoung96216 and health13399; Table S13). The latter completes our previous study in which HbTCTP transcripts were degraded by two distinct sRNAs, miRf12236-akr and miR1023b-3p (Deng et al., 2016).

Unlike reports in the literature (Chen, 2004), we found no evidence of AP2/ERFs transcripts targeted by miR172 in our experimental conditions. Instead, these transcripts were targeted by other known miRNAs (miR156, miR167, miR9386 and miR396) and by 97 newly identified miRNAs. Examples of the good quality of the detection of miRNA-mediated cleaved transcription factors (HbMYC, HbRAV-03, HbERF-VIIa8 and HbERF-VIIa5), with a degradome category of 0, in a T-plot generated by CleaveLand, can be seen in Fig. 8.

Figure 8 T-plots generated by CleaveLand pipeline for miRNA/transcription factor pairs with a degradome category 0 (many readsstarting at the cleavage site). (A) HbMYC. (B) HbRAV-03. (C) HbERF-VIIa8. (D) HbERF-VIIa5.

Identification of putative ta-siRNA-producing loci and their targets

In addition to miRNAs, another type of 21-nt small RNA may also target transcripts to induce their degradation. Trans-acting siRNAs (ta-siRNAs) are generated through the processing of a double-stranded RNA precursor, transcribed from a TAS gene targeted by a single 22-nt miRNA or by two 21-nt miRNAs (Bologna & Voinnet, 2014), through cleavage sites spaced at 21-nt intervals along its sequence (Komiya, 2017). Putative Hevea loci producing ta-siRNAs were identified by mapping sequenced 21-nt sRNAs on the assembled PB 260 genomic sequences (Fig. 3). This step led to the localisation of 88 non-redundant putative loci with the ta-siRNA originating from the latex of healthy mature trees, none from TPD-affected mature trees, and 12 from juvenile plants (Table S14, Fig. 3).

Annotation of the loci showed that two loci correspond to sequences coding for calcium-dependent protein kinase 1 (CDPK1) gene, and that 46 match the intron of farnesyl diphosphate synthase (FINT1), a gene containing a transposon (Zhang, Leclercq & Montoro, 2017). Finally, up to four ta-siRNA-producing loci were found in repeat-rich regions such as putative retrotransposons () and unannotated genomic scaffolds (0 to 38) (Table S14). No known TAS gene was identified in the Hevea genomic sequence. Degradome analysis showed that the number of cleaved targets (15) was lower than miRNAs (Fig. 3, Table 3). Most ta-siRNAs cleaved fewer targets than miRNAs. Most of these targets were found in the leaf (6) whereas only one target was found in latex (Table 3 and Table S15). An important accumulation of cleavage products was observed for the CL7Contig17 transcript, which could point to highly efficient ta-siRNA-mediated processing in the latex of TPD-affected trees (accession TPD_latex_scaffold0858_100404). No targets corresponding to ta-siRNAs derived from coding sequences were identified.

Table 3 Identified targets cleaved by ta-siRNAs in the tissue-specific transcriptomes and their functional annotation by GO.

Tissue-specific transcriptome/ Degradome libraries	Number of targets	Biological process	
Latex	12	Carbohydrate metabolic process, response to stress, transmembrane transport	
Root	1	–	
Leaf	6	Cell morphogenesis, nucleoside metabolic process, ATP synthesis coupled proton transport, ATP hydrolysis coupled proton transport, oxidation–reduction process	
Bark	4	Translation, clathrin coat assembly	
Embryos	1	–	
Reproductive tissues	1	–	

Annotation of the ta-siRNA targets (Fig. 2, Table 3 and Table S16) showed that these targets were either involved in different biological processes, linked to those in which miRNA targets played a role, or that their mode of post-transcriptional regulation differed, depending on the nature of the tissue. For instance, most transcripts involved in sugar metabolism and transmembrane transport were regulated by ta-siRNAs in latex, while in bark, they were regulated by miRNA (Table 3, Tables S15 and S16). Moreover, compared to miRNA targets, more ta-siRNA targets were involved in rubber synthesis, hormone signalling and laticifer differentiation (HbREF8, HbHMGS1, JAZ6, Topless, MED25, CTR2, and HbMYB gene families; Tables S17 to S21). Overall, most of the targets cleaved by ta-siRNAs differed from those cleaved by miRNAs. A target cleaved by both types of sRNA required two distinct sites.

Figure 9 Phylogenetic tree for Hevea and other plants of AGO andDCL proteins.

The Hevea amino acid sequences were aligned with the full-length amino acid sequences of genes in Populus trichocarpa Ricinus communis and Arabidopsis thaliana to construct a phylogenetic tree by MrBayes (v3.2.6 x64). (A) AGO proteins. (B) DCL proteins.

Identification and regulation of genes involved in microRNA biogenesis and regulatory functions in latex cells

MiRNA production requires MIR gene multi-step processing, which involves several multigene families. To annotate them in Hevea, 27 protein sequences involved in miRNA biogenesis and activity in Arabidopsis (Naqvi et al., 2012) were used as queries. Among them, 24 unique loci encoding putative orthologues were identified in the Hevea reference genome (Tang et al., 2016) using a sequence similarity search, including members of the AGO, DCL, HST, HEN1, SDN and SE families. Analysis of AGO and DCL gene families showed that each of them included four paralogous clades, in accordance with previous reports (Fang & Qi, 2016; Fukudome & Fukuhara, 2017). Compared to other plant genomes, Hevea brasiliensis exhibits several AGO genes (13, including five putative AGO5 orthologues and two putative AGO6 orthologues; Fig. 9A, Fig. S1), comparable in size to that of the recently duplicated Populus trichocarpa genome (15 genes including three AGO5 and one AGO6) (Lei et al., 2012). This means this gene family is bigger and more diversified than in the genomes of Ricinus communis (eight AGO genes) and Arabidopsis thaliana (10 genes). It was difficult to assign AGO2/3 (two genes found in Hevea: mRNA00000107 and mRNA00000478) and AGO4/8/9 (only one gene found in Hevea: mRNA00000007). By contrast, the DCL gene family of Hevea (five genes, including two putative DCL2 orthologues; Fig. 9B, Fig. S2) was in the same size range as that of both Ricinus communis and Arabidopsis thaliana (four genes each), while being smaller than that of Populus trichocarpa (six genes including two DCL2 orthologues).

The transcripts of the putative Hevea DCL1 and AGO1 orthologues (mRNA00000029 and mRNA00000323, respectively; Fig. 10) were identified, and their expression response in response to ethephon stimulation and TPD occurrence was analysed (Montoro et al., 2018) (Table S22). DCL1 expression was significantly downregulated as a result of ethephon treatment (1,591 vs. 659 reads), but not according to TPD. A very high level of expression was observed for AGO1 across all conditions tested (>19,000 reads). While putative members of the Serrate, Hasty and SDN gene families were found to have variable expression levels depending on the family member, no evidence was found for regulation in either stressed or TPD conditions (Fig. 10, Table S22). In latex, putative orthologous transcripts were detected for each gene involved in either miRNA biogenesis or its regulation. This suggests that the components of the miRNA machinery are functional in this tissue. However, whether these different genes play the same roles in rubber as in model plant species remains to be determined. Indeed, Hevea AGO and DCL gene families likely expanded due to whole genome duplication (Deng et al., 2016).

Figure 10 Analysis of miRNA biogenesis in Hevea brasiliensis.

Transcripts encoding miRNA biogenesis genes were identified by BLASTX analysis [DCL: Ribonuclease III-like DICER; SE: SERRATE (SE = mRNA00000153; SE-X2 = mRNA00000144-X2); HYL: Hyponastic leaves; AGO: ARGONAUTE; SDN: Small RNA degrading nuclease (SDN.1 = mRNA00000175; SDN.2 = mRNA00000202 and SDN.3 = mRNA00000068); HST: HASTY (HST = mRNA00000162; HST variant X2 = mRNA00000150-X2); HEN: Hua enhancer].

Discussion

The aim of this study was to identify MIR genes and their corresponding miRNAs in genomic sequences of Hevea clone PB 260. In the second step, the targets of these miRNAs were identified by analysing the degradome in different tissues. In a previous study (Gébelin et al., 2012), we described a surge in the accumulation of 21-nt sRNAs in the latex of TPD-affected rubber trees. Thanks to our miRNA annotation, we now show that these 21-nt sRNAs were not miRNAs derived from MIR genes. Therefore, miRNAs do not contribute to the 21 nt surge in sRNAs observed in TPD-affected trees. These 21 nt sRNAs could be classified as siRNAs, and may correspond to epigenetically-activated siRNA (easiRNA) (Borges & Martienssen, 2015). These siRNAs may be responsible for target transcript degradation through a pathway that is not directly dependent on miRNA-based cleavage (Bologna & Voinnet, 2014). This result underlines the relevance of analysing Hevea sRNAs beyond only the miRNA-based regulation pathways, in particular those produced by transposable elements.

Degradome analyses are generally performed with pooled tissue samples or with a single tissue. The present work highlighted tissue-specific regulation thanks to the parallel analysis of six tissue degradome libraries. Negative co-regulation of miRNA expression and its target is an important step in demonstrating the post-transcriptional control mechanism. For example, in response to saline stress, cleavage of chloroplastic CuZnSOD transcripts was correlated with upregulation of miRNA398 expression in bark and in roots (Gébelin et al., 2013a), but not in leaves. This regulation was not found in relation with the other stresses tested (drought, wounding, flooding, cold, high light intensity (Gébelin et al., 2013a)). In the degradome analysis, the target chloroplastic CuZnSOD was identified only in roots and bark with a small number of transcripts at the cleavage site, in perfect agreement with the results of co-regulation analyses. Although preliminary, our study enabled us to validate the link between degradome analysis and co-regulation analysis. It also underlines the need to account for the spatio-temporal dimension as well as the physiological context involving a case-by-case study.

We also demonstrated the partial conservation of sRNA machinery and targets in Hevea. Conservation of miRNA-target pairs between plant lineages is well described (Cuperus, Fahlgren & Carrington, 2011). However, in a previous study, we showed partial conservation between Hevea and Arabidopsis miRNA binding sites for genes encoding ROS-scavenging enzymes (Gébelin et al., 2012; Zhang et al., 2019). Another example of partial conservation between plant is the targeting of AP2/ERF transcripts by miR172, which is well-documented in different plant species (Aukerman & Sakai, 2003; Wang et al., 2017b), but in the present study, we did not observe such regulation in any of the well-characterised AP2/ERF transcripts in Hevea (Piyatrakul et al., 2012; Putranto et al., 2015a).

MiRNA-guided cleavage of TAS transcripts, which triggers the subsequent production of ta-siRNA of 21-nt increments, is the standard model of ta-siRNA production established in Arabidopsis. Our data suggest that considerable ta-siRNA production occurred in Hevea in conjunction with post-transcriptional cleavage of their target transcripts, whereas no TAS genes were identified in the rubber genome sequence. Other authors have proposed that, even in Arabidopsis, concurrent mechanisms may exist for ta-siRNA production (Yu et al., 2018; Yu et al., 2015).

Figure 11 A simplified natural rubber biosynthesis pathway showing the miRNA cleaving the target transcripts.

miRNAs found only in latex are shown in bold.

Conclusion

This study provided several new insights into Hevea concerning miRNA biogenesis and activity, tissue-specific post-transcriptional regulation in response to abiotic stress, and the impact of Hevea speciation with the whole genome duplication shared with cassava. The divergence observed from annual plant species suggests enhanced adaptation to stress in Hevea, particularly in laticifers, which are subjected to recurrent combinatorial stresses. Concerning the post-transcriptional regulation observed for sucrose metabolism in bark and in each step of the natural rubber biosynthesis pathway (Fig. 11), intra- and interspecific comparison of rubber-producing plants is needed to decipher the impact of post-transcriptional regulations in response to stress in the context of rubber production.

Supplemental Information

Table S1 Degradome library cleaning process.

Click here for additional data file.

Table S2 MITP annotation of 5 small RNA–seq data forclone PB 260

Small RNA from the latex of healthy and TPD-affected trees, from juvenile plants and from mature and young leaves. Filtered_reads: t he reads which were not mapped to mRNA, rRNA, tRNA and known miRNA. Cluster_number: the number of reads cluster which was meant to get candidate regions for miRNA. Cluster_reads: all the reads number which is the sum number of each cluster. Extracted_seq: the number of sequences which was used to create precursor and mature sequence and gff file for candidate clusters. Extracted_seq_after_filter: the number of sequences which was filtered based on candidate miRNA (hairpin) sequence and structure attribute values. Extracted_seq_unique: the number of sequences which was non-redundant hairpin according to the genome position. Candidate_attribute_filter: the number of sequences which was filtered by attribute values such as default MEF (minimal folding free energy) and AMFE (adjusted mfe) with non-redundant sequence.

Click here for additional data file.

Table S3 Sequence file of 1042 unique hairpin structures producing sRNAs in clone PB 260 and identified by the MITP program

Click here for additional data file.

Table S4 Annotation by Blast of hairpins sharing homologies with known sequences in the MirBase and PMRD databases

Click here for additional data file.

Table S5 Sequence file of 1839 small RNA sequences of 20-22 nt from the hairpin structure, with or without known homology

Click here for additional data file.

Table S6 CleaveLand output for three experimentally validated targets (Gébelin et al., 2012)

These pairs are SBP/miR156, ARF/miR160 and chloroplastic CuZnSOD /miR398.

Click here for additional data file.

Table S7 CleaveLand output using tissue-specific transcriptome databases, sRNAome of 20 to 22 nt and tissue-specific degradome libraries

Click here for additional data file.

Table S8 Functional annotation of target cleaved by miRNA by Gene Ontology

Click here for additional data file.

Table S9 CleaveLand output using a list of genes involved in natural rubber biosynthesis, sRNAome of 20 to 22 nt and tissue-specific degradome libraries

Click here for additional data file.

Table S10 CleaveLand output using a list of gene encoding AP2/ERF members, sRNAome of 20 to 22 nt and tissue-specific degradome libraries

Click here for additional data file.

Table S11 CleaveLand output using a list of genes involved in the JA signalling pathway, sRNAome of 20 to 22 nt and tissue-specific degradome libraries

Click here for additional data file.

Table S12 CleaveLand output using a list of genes involved in the ET biosynthesis and signalling pathways, sRNAome of 20 to 22 nt and tissue-specific degradome libraries

Click here for additional data file.

Table S13 CleaveLand output using a list of genes regulated during TPD occurrence, sRNAome of 20 to 22 nt and tissue-specific degradome libraries

Click here for additional data file.

Table S14 Annotation of ta-siRNA producing loci by blast and according to the sRNA library

Click here for additional data file.

Table S15 CleaveLand output using tissue-specific transcriptome databases, phase siRNA of 21 nt and tissue-specific degradome libraries

Click here for additional data file.

Table S16 Functional annotation of targets cleaved by ta-siRNA by Gene Ontology

Click here for additional data file.

Table S17 CleaveLand output using a list of genes involved in natural rubber biosynthesis, ta-siRNA of 21 nt and tissue-specific degradome libraries

Click here for additional data file.

Table S18 CleaveLand output using a list of gene encoding AP2/ERF members, ta-siRNA of 20 to 22 nt and tissue-specific degradome libraries

Click here for additional data file.

Table S19 CleaveLand output using a list of genes involved in the JA signalling pathway, ta-siRNA of 21 nt and tissue-specific degradome libraries

Click here for additional data file.

Table S20 CleaveLand output using a list of genes involved in the ET biosynthesis and signalling pathways, ta-siRNA of 21nt and tissue-specific degradome libraries

Click here for additional data file.

Table S21 CleaveLand output using a list of genes regulated during TPD occurrence, ta-siRNA of 21 nt and tissue-specific degradome libraries

Click here for additional data file.

Table S22 RNA-seq analysis of genes involved in miRNA biogenesis (count and EdgeR)

Click here for additional data file.

Figure S1 Alignment of AGO sequences for phylogenetic analysis

Click here for additional data file.

Figure S2 Alignment of DCL sequences for phylogenetic analysis

Click here for additional data file.

Supplemental Information 1 Latex-specific transcriptome

Consensus sequences

Click here for additional data file.

Supplemental Information 2 Leaf-specific transcriptome

Consensus sequences

Click here for additional data file.

Supplemental Information 3 Embryo-sepecific transcriptome

Consensus sequences

Click here for additional data file.

Supplemental Information 4 bark-specific transcriptome

Consensus sequences

Click here for additional data file.

Supplemental Information 5 Flower-specific transcriptome (male and female)

Consensus sequences

Click here for additional data file.

Supplemental Information 6 Root-specific transcriptome

Consensus sequences

Click here for additional data file.

Supplemental Information 7 Degradome sequencing from embryo

Raw data from Illumina-seq

Click here for additional data file.

Supplemental Information 8 Degradome sequencing from bark

Raw data from miSeq

Click here for additional data file.

Supplemental Information 9 Degradome sequencing from root

Raw data from miSeq

Click here for additional data file.

Supplemental Information 10 Degradome seqeuncing from leaf

Raw data from miSeq

Click here for additional data file.

Supplemental Information 11 Degradome sequencing from flower

Raw data from MiSeq

Click here for additional data file.

Supplemental Information 12 Degradome data from latex

Raw data from MiSeq

Click here for additional data file.

The authors would like to thank Zheng Li, the CIRAD-INRA representative in Beijing, and Estelle Jaligot for careful reading, comments and suggestions on the manuscript. This work was supported by the CIRAD—UMR AGAP HPC Data Center of the South Green Bioinformatics platform (http://www.southgreen.fr/). We are most grateful to the GPTR (Great regional technical platform) core facility for its technical support. JL and PM conceptualized the research program. JL designed the experiments. B.Far. and HV generated the degradome data. SW performed most of the data analyses with the help of SG, SP and B.Fav. K supervised the field experiments. J-FD performed the phylogenetic analyses. SH and CT provided the reference Hevea genome sequence. EO-A and GD worked on PB260 re-sequencing data. JL and PM wrote the manuscript. All authors discussed results and commented on the manuscript.

Additional Information and Declarations

Competing Interests

Author Contributions

DNA Deposition

Data Availability

The authors declare there are no competing interests.

Julie Leclercq conceived and designed the experiments, prepared figures and/or tables, authored or reviewed drafts of the paper, and approved the final draft.

Shuangyang Wu conceived and designed the experiments, analyzed the data, prepared figures and/or tables, authored or reviewed drafts of the paper, and approved the final draft.

Benoît Farinas, Hélène Vignes and Kuswanhadi Kuswanhadi performed the experiments, prepared figures and/or tables, and approved the final draft.

Stéphanie Pointet, Bénédicte Favreau, Shenghan Gao and Gaëtan Droc analyzed the data, prepared figures and/or tables, and approved the final draft.

Enrique Ortega-Abboud, Songnian Hu and Chaorong Tang analyzed the data, authored or reviewed drafts of the paper, and approved the final draft.

Jean-François Dufayard analyzed the data, prepared figures and/or tables, authored or reviewed drafts of the paper, and approved the final draft.

Pascal Montoro conceived and designed the experiments, authored or reviewed drafts of the paper, and approved the final draft.

The following information was supplied regarding the deposition of DNA sequences:

Data is available at the GSA and BIG Data Center: PRJCA001333.

http://bigd.big.ac.cn/gsa/s/k13V462b.

SmallRNA-seq from latex of healthy trees:

Leclercq, Julie (2020): MS_1_sequence_plus.zip. figshare. Dataset. https://doi.org/10.6084/m9.figshare.10264979.v1.

SmallRNA-seq from latex of tapping panel affected trees:

Leclercq, Julie (2020): MS_8_sequence.zip. figshare. Dataset. https://doi.org/10.6084/m9.figshare.10264985.v1.

SmallRNA-seq from lyooung plantlet (stem, root and leaf):

Leclercq, Julie (2020): MS_1_sequence_plus.zip. figshare. Dataset. https://doi.org/10.6084/m9.figshare.10264979.v1.

The following information was supplied regarding data availability:

Degradome data and tissue-transcriptome data are available at the BIG Data Center Genome Sequence Archive: subCRA001506.

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
