# Peer review of "Post-transcriptional regulation of several biological processes involved in latex production in Hevea brasiliensis"

_PeerJ, doi:10.7717/peerj.8932_

## Round 0.1 · original submission · Major Revisions

There appear to be some misdirection in presenting the key RNA species which need study. Reviewers see merit in the study but require some clarity and background information to make the findings more accessible to the common reader. Please take a look at the reviewers notes and attempt to better address their concerns. It may require a simpler breakdown of the RNA species to help differentiate the RNA forms that are being detailed. The status of the manuscript is currently at the Major Revision state until some clarification can be presented. I agree that the work has merit and may require some re-organization for the way it is organized, discussed, and presented. Thank you for your contribution.

Reviewer 1 ·

Basic reporting

Minor concerns

Abstract: the authors should cite the importance of small RNAs during viral infections (lines 29-31)

Abstract: Authors should avoid contractions in the abstract before define them, such as MIR and ta-siRNAs (line 36)


Line 146: Table 1 is linked to a publication (Zhang et al., 2019) which is not present in the reference list. Since there is not publication accessible, authors could explain why use bowtie instead bowtie2 to the alignment of paired-end long reads?

In Figure 11, authors should highlight Hevea proteins to improve the understanding. In the case of the other organism proteins, they should be named according to its gene symbol no only accession code.


Main concerns

One of the main characteristics of miRNAs is the base preference. Most of miRNA show 5’ U base preference. The authors should show base preference per size for small RNAs derived from miRNAs as a positive control

Texts within images in some main figures (1,2,5, 6, 8). They need to be improved to allow better interpretation of the images.

The authors produced a large number of images, which I believe the number is exaggerated in the case of main figures, but do not produce an image demonstrating Biological Processes of cleaved targets? Why do not perform GO enrichment analysis?

Figures 9 and 10 are not legible nor understandable. I suggest to improve them, enlarge in supplemental figures or remove.

Experimental design

no comment

Validity of the findings

- I believe the authors showed a large amount of data to prove its hypothesis. However, some basic analysis should be performed in order to corroborate their findings:

- Describe about the identification of highly conserved miRNAs (such as let-7 and others)
- Base preference od miRNA-derived small RNAs
- Correlation between miRNA expression, transcript expression and degradome, demonstrating the complete mechanism of regulation (since we expects high expression of miRNAs leading to decreasing in the expression of mRNAs due its degradation assessed by degradome)

Additional comments

In this manuscript, Leclercq & Wu et al took advantage of RNA deep sequencing to assess the transcriptional profile of Hevea brasiliensis and its regulation strategies. The authors made an incredible effort to integrate a large number of datasets. Although I believe the manuscript is well written and full of data, the manuscript in very dense and needs simplification and clarification in some parts. Therefore,
there are some points should be addressed in order to improve the reading and understanding of the findings.

Reviewer 2 ·

Basic reporting

The authors present a detailed analysis of small RNA populations, degradome, and RNA expression in Hevea to better understand the potential changes in trees with Tapping Panel Dryness (TPD). Analyses further define the molecular changes and functional networks in TPD affected trees. In addition, data is provided for multiple Hevea tissues types. The authors find that the increase previously reported for 21-nt siRNAs is not due to miRNA increase, but rather siRNAs.

While not clearly stated, it appears that a driving hypothesis behind this research was investigation of post transcriptional regulation associated with Tapping Panel Dryness (TPD). Previous research has shown a change in small RNA size (lines 81-82). Unfortunately, this comes off as a side bar in the introduction and results. The research question being addressed needs more clarification and addressed in these sections. Otherwise this manuscript comes off as more observational. It is an interesting finding that 21-nt siRNAs increase in TPD. Unfortunately, the focus of the manuscript is on miRNAs and impacted pathways, rather than genes that might be impacted by siRNAs in the disease response. This would be an interesting analysis to include.

The figures and tables presented are challenging to interpret and make the manuscript inaccessible to general audiences. The authors should revise the figures that can be interpreted by audiences not familiar with the complex statistics behind the analysis. For example, I could not follow the analysis of cleaved targets described (lines 193-201), and relate this to the results presented (Figures 6, 8). Tables 1, 2, 3, 4, 7, 9 do not help explain the results found. They are simply descriptive statistics. Figures 9 and 10 are illegible; I cannot tell what the networks described here would be.

Please include a description of the data provided in all supplemental materials.

Overall, this manuscript could be greatly improved through focus on a specific question and in depth analysis. The figures need to be redone to help the reader follow the conclusions made by the authors.

Experimental design

The methods used are appropriate and follow rigorous statistical analyses.

Validity of the findings

I don’t see a significant difference in small RNA populations in Figure 5, as previously reported and discussed by the authors (Line 283). This requires explanation why there might be a difference between the reports.
I find limited value in the analysis of post transcriptional gene families in Hevea. This does not add to the manuscript as written. The analysis could be in improved with a broader phylogenetic analysis and comparative genetics with other species. The manuscript could be improved by limiting or removing this component and focusing on the gene regulation in Hevea.
The authors present a case that canonical miRNA and ta-siRNA genes have evolved to target new gene families; this is quite a novel discovery as miRNA:RNA pairings have been shown to be highly conserved throughout plant evolution (starting line 327). A deeper analysis, showing lack of sequence homology with canonical miRNA:RNA pairs is required to make this strong conclusion. Very strong conclusions are made (350-354) that need to be supported with sequence alignments and clear demonstration of cleavage. The authors are suggested to review mIRNA literature for published ways to show this information.
The link between the ethephon treatment and TPD plants was not clear to me. These appear to be disparate experiments pulled together in one paper. The authors should tie these analyses together, if they should be (405-423).

---

## Round 0.2 · accepted · Accept

Thank you for addressing many of the concerns provided from the reviews. I believe you have addressed the concerns adequately and will forward the manuscript for further consideration as it looks acceptable. Usually Venn diagrams do not have much to offer; however, yours appeared quite interesting and helped define specificity. I believe this manuscript may offer some new insights into regulatory networks; I congratulate you on your efforts.